# Task and Ego Orientation in Sport Questionnaire (TEOSQ): Psychometric Properties in Its Digital Version

**DOI:** 10.3390/ijerph19063251

**Published:** 2022-03-10

**Authors:** Verónica Morales-Sánchez, Nuria Pérez-Romero, María Auxiliadora Franquelo, Isabel Balaguer, Antonio Hernández-Mendo, Rafael E. Reigal

**Affiliations:** 1Department of Social Psychology, Social Work, Social Anthropology and East Asian Studies, University of Málaga, 29016 Málaga, Spain; vomorales@uma.es (V.M.-S.); nuriapr97@uma.es (N.P.-R.); auxifranq@gmail.com (M.A.F.); mendo@uma.es (A.H.-M.); 2Department of Social Psychology, University of Valencia, 46010 Valencia, Spain; isabel.balaguer@uv.es

**Keywords:** achievement goals, sport, invariance, psychometric properties

## Abstract

The Task and Ego Orientation in Sport Questionnaire’s (TEOSQ’s) psychometric properties have been explored in previous studies but never in its digital version, which facilitates data collection. The objective of this study was to analyze the psychometric properties of the online TEOSQ by MenPas 1.0. The sample was composed of 2320 users (58.4% women; 41.6% men), between 18 and 65 years old (M = 25.27, SD = 7.39). The methods used were Confirmatory Factor Analysis (CFA) and invariance analysis from the original 13-item model. The CFA was corrected for all samples (CFI = 0.92–0.94, TLI = 0.93–0.95, RMSEA = 0.07, SRMR = 0.06; df = 64; Bollen-Stine Bootstrap p = 0.02–0.07): general sample (χ^2^ = 720.72, χ^2^/df = 11.26), women (χ^2^ = 496.85, χ^2^/df = 7.76), men (χ^2^ = 321.67, χ^2^/df = 5.03), individual sports (χ^2^ = 525.26, χ^2^/df = 8.21), and team sports (χ^2^ = 306.01, χ^2^/df = 4.78). The results also indicate optimal adjustments for invariance: convergent, discriminant and composite reliability validity. The study evidence demonstrates the adequate psychometric properties of the digital version. To conclude, considering the results obtained, the model shows a good psychometric fit for the sample in its online format. The principal limitations were computer failures, as well as that the large part of the sample were users between 20 and 25 years old, so the heterogeneity should be improved. The practical implications of this study could improve the efficacy of data collection in sports motivation using the online TEOSQ.

## 1. Introduction

Duda and Whitehead [1] presented the differences and similarities in the conceptualization of achievement goals made over the years by Nicholls [2,3], Dweck [4], and Ames [5]. They all agreed that one central concept in achievement goal framework is the concept of competence. These theorists in the field of educational psychology agree that there are at least two ways of judging competence in achievement contexts: one self-referential or mastery-centered (meta-task) and the other comparative or normative (meta-ego). Both also consider that variations in achievement goals are related to cognition, affects, and behaviors.

Following Duda and Nicholls [6], task-oriented goal dimension focuses on improving one’s own ability, acquiring knowledge, and the belief that it is necessary to put forth effort, try to understand tasks, and collaborate with peers in order to succeed. In addition, the ego-orientation dimension is characterized by the desire to outperform others and the belief that success requires greater ability relative to an external criterion. Although the two dispositional goal orientations have some stability, they are viewed not as traits [1], but as cognitive schemas subject to change [2]. Nicholls suggests that in every achievement activity, people will be guided by their personal goal orientations and that these personal goals, also known as “motivational orientations”, reflect the criteria people use to judge success [2]. To assess whether people are predominantly guided by task or ego orientation when performing an activity, Duda [7] developed the Task and Ego Orientation in Sport Questionnaire (TEOSQ).

In the context sports, it has been observed that task-oriented individuals try to improve their skills, have greater persistence, cooperate with others, and make more effort to improve their skills. However, those who are ego-oriented seek to demonstrate superior competence during sport practice, or surpass normative standards in the sport. Ego-oriented athletes are more likely to develop addictive behaviors such as substance abuse or dropout in sport [8,9,10,11,12].

The TEOSQ has been used on many occasions, in which is related to beliefs about the causes of success in sports, enjoyment, interest, affect, aggressive and fair play attitudes, learning, anxiety, and coping strategies, among others [1]. This instrument has been widely used in the context of sports, both in English and in other languages [1]. Specifically, the TEOSQ has been among the six most cited questionnaires with respect to the assessment of motivation in sports [13]. This questionnaire has shown its validity, reliability, and internal consistency on several occasions, for different ages as well as cross-culturally [14,15,16,17]. In addition, the cross-cultural invariance of its psychometric properties has also been highlighted in Mexican and Spanish young athletes (aged 11 to 18 years) [16] and in adolescents from Spain, Argentina, Colombia, and Ecuador [14].

One of the most novel aspects of the present study is that the questionnaire is completed online. Digital platforms have several advantages such as the possibility of covering more participants and a higher diversity in samples, faster and cheaper coverage, participants from different places, and without an evaluator [18,19,20]. This also increases the use of analysis techniques that require a large sample and allows the faster validation of instruments [21]. However, this type of methodology also presents limitations as there is lower control of variables in the evaluation [19,20]. Therefore, many implications should be investigated as they may affect reliability [19].

An example of this can be found in González-Ruiz et al. [19] in which anonymity increased reliability by decreasing social desirability. In addition, Raimundi et al. [22] evaluated the Argentine population in a written format and in an online format, obtaining differences in the levels of validity and reliability. They found higher internal consistency indices in the sample that performed the online questionnaire on certain scales (Visualization Control, Motivational Level, Positive Coping Control, and Attitudinal Control), while they found higher internal consistency indices for the paper version in Self-Confidence on the Negative Coping Control scale. Then, they raised future lines of questions such as the influence of the presence of the evaluator, anonymity, and the conditions of the evaluation. In addition, online platforms allow the use of new variables that can also influence the reliability of the instruments, among which are the response time to the items, the time needed to complete the questionnaire, the order in which the questionnaire is answered, or even the modifications of answers [19,21]. These new forms of interaction may be surrounded by distractors and stimuli that differ from the online assessment and should continue to be studied to show evidence [21].

In addition, events may arise that further impair this work, such as the current situation caused by COVID-19, in which we have found it difficult to gather large groups of people in person. In recent years, the use of virtual media for data collection has increased [23,24,25]. Because of that, it is interesting to analyze not only the psychometric properties of a questionnaire, but also the nature of the tool used. Among the set of digital alternatives available, one of them is MenPas 1.0 Psychosocial Assessment Platform (www.menpas.com, accessed on 10 March 2021), which allows both athletes and organizations to be assessed quickly, obtaining the results immediately, and, in addition, allows large databases to be collected to help in the evaluation of instruments [18,20].

Finally, another interesting aspect of the psychometric analysis is the analysis of invariance, as it indicates whether the results of different types of samples can be compared [26,27]. However, since this questionnaire has only been conducted with a sample of young people, it could be interesting to further explore other population groups, such as the adult population, and this is precisely one of the reasons for the present study. It is also interesting to continue exploring the differences in goal orientations as a function of gender, since some studies have reported that girls were more task-oriented than boys [8,15]. On the other hand, in the case of the type of sport, in the systematic review conducted by Lochbaum et al. [28], it was indicated that there are not enough data to state that one type of sport (team or individual) promotes a higher level of ego climate or task. However, none of the studies performed invariance analyses to test whether groups according to gender or according to the type of sport can be compared. In this way, one of the objectives of the present study is to explore invariance as a function of gender and type of sport.

Therefore, the main objective of this study was to analyze the psychometric properties of a digital version of the TEOSQ by performing confirmatory factor analysis, estimating the internal consistency, convergent validity, discriminant validity, as well as the invariance of the measure according to gender and type of sport.

## 2. Materials and Methods

### 2.1. Participants

The study covered 2320 users of the MenPas Psychosocial Assessment platform (www.menpas.com, accessed on 10 March 2021), aged between 18 and 65 years (*M* = 25.27, *SD* = 7.39). Of the total sample, 58.4% were female (*n* = 1355) and 41.6% male (*n* = 965). A total of 64.8% of participants practiced individual sports (*n* = 1503) and 35.2% team sports (*n* = 965). In the total sample, there were 63 different sports, of which 21 were individual, 15 were team, and 27 were adversarial. Of the total (*n* = 2320), 82.2% had received a higher education (*n* = 1908), 15.1% had received an intermediate education (*n* = 351), 2.2% had received a primary education (*n* = 50), and 0.5% (*n* = 11) had received no education.

### 2.2. Instrument

The instrument used was the Task and Ego Orientation in Sport Questionnaire (TEOSQ) [7], specifically a computerized version of the questionnaire in Spanish [8], which can be found in MenPas 1.0. The questionnaire consists of 13 items divided into two scales that assess Task Orientation (7 items) and Ego Orientation (6 items). The instructions ask athletes to think about when they feel most successful in their sport by asking: “I feel most successful in my sport when…”. Responses are collected using a 5-point Likert-type scale ranging from (1) strongly disagree to (5) strongly agree. An example of a task orientation item would be “I learn a new skill by trying very hard” and an Ego Orientation item would be “I am the best”.

Psychometric analysis of the Spanish version of the questionnaire [8] offered adequate adjustments after orthogonal rotation, in which of the 49.2% of the total variance, 25.1% was explained by the task orientation factor and the remaining 24.1% by the ego orientation factor. The internal consistency was carried out with Cronbach’s Alpha, obtaining 0.78 for task orientation and 0.80 for ego orientation.

### 2.3. Procedure

First, the TEOSQ questionnaire was implemented in the MenPas 1.0 Psychosocial Platform (www.menpas.com. accessed on 10 March 2021) from its Spanish version [8]. The sample data were collected through this platform [18,20] during the dates between 13 February 2011, and 10 March 2021. Through this platform, participants had to register as users by filling in their sociodemographic data (name, sport practiced, gender, age, studies, and profession among others), although not all sociodemographic data were necessary in the study. Once inside the platform, they completed the TEOSQ. In addition, the ethical principles of the Declaration of Helsinki ([29] and the Oviedo Convention [30] were respected throughout the research process. The work was approved by the Ethics Committee of the University of Malaga Number 19-2015-H.

### 2.4. Data Analysis

The descriptive statistics of the sample were carried out. Confirmatory Factor Analysis was performed based on the structure available in the literature reviewed [8]. A Maximum Likelihood (ML) procedure was used as the estimation method. The model fit was performed through Chi-square (χ^2^), degrees of freedom (df), and significance level. The Comparative Fit Index (CFI), Tucker–Lewis Index (TLI), Root Mean Square Error of Approximation (RMSEA) with a confidence interval (90% CI), and Standardized Root Mean Residual (SRMR) were also calculated. Following the literature [31,32,33,34] for CFI and TLI, the cut-off point was ≥0.90 and for RMSEA and SRMR was ≤0.08. In addition, convergent validity was calculated using the average variance extracted (AVE) ≥ 0.50, and discriminant validity was calculated using the AVE of each latent variable, this being greater than the square of the correlation between them. AVE is a complementary measure of composite reliability, where ρ > 0.50 means that a high percentage of variance is explained for the construct compared to the variance of the measurement error. Internal Consistency was also calculated by estimating the composite reliability index, with a 0.70 cut-off [32]. These analyses were carried out through two statistical programs, IBM SPSS Statistics, Version 23, and AMOS 23 (BM, Inc., Chicago, IL, USA).

### 2.5. Invariance Analysis

The invariance analysis was performed to examine the constancy of psychometric properties between different groups [26,31]. The model must be adjusted for each group, and it is necessary to examine the four types of invariance. The four types are configural (implies that the same item must be associated with the same factor in each group), metric (compares regression slopes or score changes), scalar (indicates that the scores of the different groups have the same unit of measurement and the same origin), and residual (group differences in the items are only due to differences in the factors). To be able to state that a model is invariant, following Chen [35], there must be changes of ≤0.01 for CFI (ΔCFI), changes of ≤0.015 for RMSEA (ΔRMSEA), and changes ≤ 0.030 for SRMR (ΔSRMR). Analyses were performed using the AMOS 23.0 statistical program.

## 3. Results

### 3.1. Preliminary Analysis

Through preliminary analyses, no missing data or outliers were found in the overall sample (Mahalanobis distance = p1 = 0.000; p2 = 0.000) [31]. No normality issues were found in the distribution of the samples, as the data for skewness and kurtosis ranged from ±2 to ±7, respectively [31]. However, the multivariate Mardia coefficient showed a violation of the multivariate distribution in the sample (>5.0), so a 2000-sample Bootstrap Bollen–Stime was performed [36].

### 3.2. Internal Consistency, Convergent and Discriminant Validity

The Table 1 presents the descriptive statistics, the Average Variance Extracted (AVE), and the correlation between the different factors. There were good values of average variance extracted (AVE), considered as convergent validity [32,34]. Additionally, good results are presented for discriminant validity, since the AVE for each factor is higher than the square of the correlation (*r* = 0.10; *r*^2^ = 0.01).

Table 2 shows the composite reliability, presenting adequate values for each of the samples (≥0.70). It also shows the factorial weights for each of the questionnaire items, all of which were significant.

### 3.3. Confirmatory Factor Analysis

A Confirmatory Factor Analysis was performed to analyze the structure of the questionnaire proposed by Balaguer et al. [8]. The factor analysis developed confirms a bifactor structure, consisting of 13 items: 7 items for the task factor and 6 items for the ego factor. This model (Figure 1) showed seven items for the Task variable (items 2, 5, 7, 8, 10, 12 and 13) and six items for the Ego variable (items 1, 3, 4, 6, 9 and 11). As shown in Table 3, the analysis showed an adequate fit according to the criteria shown in the literature [31,32,33] and specified in the Section 2. These analyses also showed satisfactory results for the groups separately (female, male, individual, and team). As complementary information, the results also show Confirmatory Factor Analysis performed in previous studies [14,15,17,37]. In this, it can be observed that the current study shows a greater number of indicators than the rest. In addition, it is the only one that presents an adult population.

### 3.4. Invariance Analysis

Following the previously discussed theoretical lines, the tested measurement model proved to be invariant across gender and sport type. Thus, as shown in Table 4, the changes given in CFI (∆CFI) were less than 0.01, the changes in SRMR (∆SRMR) were less than 0.03, and the changes given in RMSEA (∆RMSEA) were less than 0.015 [35]. For this purpose, the four types of invariances mentioned earlier in the Section 2 were analyzed [26,35].

## 4. Discussion

The main objective of the present study was to analyze the psychometric properties of the online TEOSQ by MenPas 1.0 (www.menpas.com, accesses on 10 March 2021). For this purpose, a Confirmatory Factor Analysis was performed, and internal consistency, convergent, and discriminant validity were estimated, as was the invariance of the measure as a function of gender and type of sport.

First, the results indicate that the questionnaire offers an adequate fit for the online sample [31,32,33]. The factor analysis developed confirms a bifactor structure, consisting of 13 items: 7 items for the task factor and 6 items for the ego factor. The structure of the model coincides with previous literature [8]. All items presented an adequate factorial weight and no cross-linkages were found so no item had to be eliminated. In addition, convergent and discriminant validity were adequate, indicating that the items were related to their respective factors. The internal consistency values also showed good fit [32], indicating that the questionnaire items differed between the two factors.

Second, invariance analyses were performed, which are a very important aspect in the analysis of psychometric properties [26,27], since they show whether there is variability between groups. In this case, the analyses showed that there was no variability between male and female populations. Furthermore, there was also no variability between the group that practiced individual sports and the group that practiced team sports. Therefore, according to different authors [26,31], it can be stated that (i) the model fits for each of the groups separately (female/male and individual/team) and that (ii) different levels of invariance were examined, confirming that the same items were associated with the same factors (configural invariance). The items were associated in the same way with each factor without changes in scores (metric invariance). The scores of both groups have the same unit of measurement and the same origin (scalar invariance). Finally, the differences in the items of each group are only due to the factors (residual invariance). Therefore, it can be affirmed that the model is invariant based on gender and the type of sport practiced, insofar as it meets the criteria established for the four types of invariance, including residual invariance, whose fulfillment is sometimes difficult to achieve, especially in the field of social sciences [26,31]. At last, it indicates that the questionnaire is equivalent for each group, presenting no structural differences and, therefore, that the evaluated construct does not depend on the characteristics of each of the groups [26,27]. This offers adequate estimators for the comparison of the mentioned groups that can be analyzed in future research. In addition, the results of this study reinforce previous research in which they determined gender differences once suitable indicators were obtained.

Given the limitations of this study, possible errors in the online collection of questionnaires can be considered, including the fact that people may not have answered honestly. There may also have been loss of information due to computer failures or the participant’s own abandonment. However, this is compensated for by the large amount of sample collected. Another limitation of this study is the way in which sociometric information is collected, since the Menpas 1.0 platform, among its sociodemographic data, requests the type of sport in the form of an open-ended question, leading to confusion in some cases. For example, some users indicate that they go to the gym, walk, or run, making it difficult to determine the type of sport or its intensity. This limitation could be solved by modifying the request for sociodemographic data by adding a question on the type of sport (individual, group, adversary, or not practicing) and another on an estimate of the intensity of the sporting activity. Finally, the difference in the number of people between men and women, the high percentage of participants aged between 20 and 25 compared to the other ages, and the difference in number between the individual and team sample may have affected the results, so in the future it would be interesting to perform the analyses with a more homogeneous sample. In future research, it would be interesting to continue carrying out online evaluations to compare the results obtained in the present study. It would be important to reinforce the limitations found, such as the homogeneity of the sample, to obtain better results. In addition, invariance studies could also focus on other groups, such as, for example, age (childhood, adolescence, and adulthood).

## 5. Conclusions

Considering the results obtained, the model shows a good psychometric fit for the sample in its online format, indicating that the questionnaire in its online form is a good instrument for assessing achievement goals. In addition, the invariance adjustments show that the questionnaire can be used to compare differences between men and women, as well as between individual and group sports practitioners. These results can become very important to be able to study the achievement goals of different athletes and to be able to propose training programs or action plans that favor the task orientation of athletes, since we know that task orientation promotes the development of skills through effort, social cooperation, and enjoyment of sport and reduces the abandonment of sport, anxiety, or drug use [1,8,11]. In addition, online assessment is becoming increasingly relevant nowadays due to the impossibility of gathering large numbers of people in the same space because of the risk status by the COVID-19 virus. This could decrease risk, time, and expense in both research and individual application.

## Figures and Tables

**Figure 1 ijerph-19-03251-f001:**
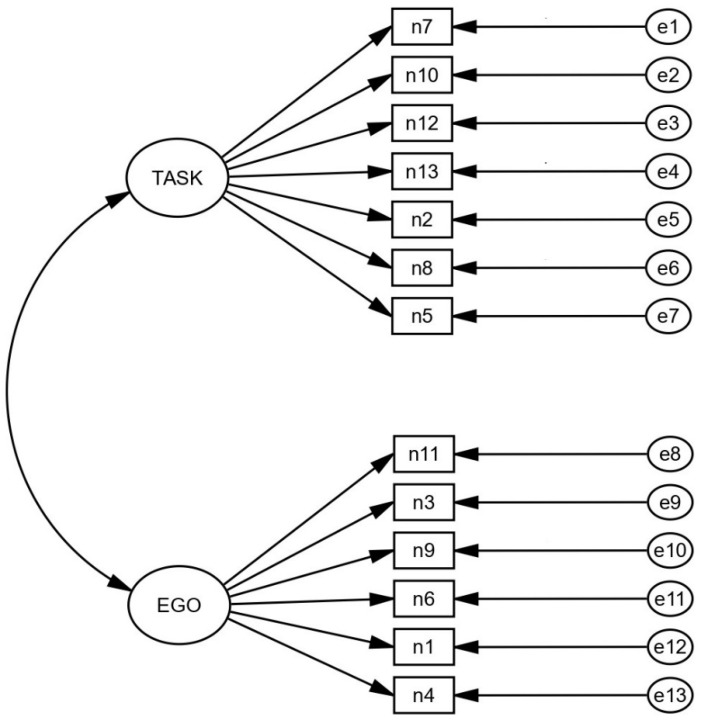
Structure of the TEOSQ.

**Table 1 ijerph-19-03251-t001:** Descriptive statistics, internal consistency, convergent validity, discriminant validity, and mean variance extracted for the general sample.

Variables	M	SD	AVE	TASK	EGO
TASK	29.28	4.38	0.47	-	
EGO	13.61	4.98	0.50	0.10 **	-

Note: M = Mean; SD = Standard Deviation; AVE = average variance extracted. ** *p* < 0.01.

**Table 2 ijerph-19-03251-t002:** Factor loadings, error and composite reliability of the TEOSQ.

	General Sample		Male Sample		Female Sample		Individual Exhibit		Group Show	
	λ	SE	λ	SE	λ	SE	Λ	SE	λ	SE
Task	0.86		0.87		0.85		0.86		0.86	
Item 2	0.77 *	0.03	0.79 *	0.04	0.75 *	0.04	0.63 *	0.05	0.60 *	0.07
Item 5	0.64 *	0.03	0.65 *	0.04	0.63 *	0.04	0.79 *	0.05	0.79 *	0.07
Item 7	0.62 *	0.03	0.63 *	0.04	0.61 *	0.04	0.77 *	0.05	0.70 *	0.07
Item 8	0.58 *	0.03	0.59 *	0.05	0.58 *	0.05	0.65 *	0.05	0.63 *	0.07
Item 10	0.79 *	0.03	0.79 *	0.04	0.79 *	0.04	0.75 *	0.05	0.80 *	0.07
Item 12	0.75 *	0.03	0.77 *	0.04	0.72 *	0.04	0.59 *	0.06	0.57 *	0.07
Item 13	0.64 *	0.03	0.66 *	0.05	0.63 *	0.04	0.64 *	0.05	0.64 *	0.07
Ego	0.86		0.85		0.85		0.85		0.87	
Item 1	0.69 *	0.03	0.70 *	0.05	0.67 *	0.05	0.72 *	0.04	0.79 *	0.04
Item 3	0.65 *	0.03	0.63 *	0.05	0.65 *	0.05	0.66 *	0.04	0.63 *	0.04
Item 4	0.66 *	0.04	0.67 *	0.05	0.64 *	0.05	0.80 *	0.04	0.80 *	0.04
Item 6	0.67 *	0.03	0.68 *	0.05	0.66 *	0.05	0.65 *	0.04	0.70 *	0.04
Item 9	0.80 *	0.04	0.80 *	0.05	0.79 *	0.05	0.68 *	0.04	0.71 *	0.04
Item 11	0.75 *	0.04	0.73 *	0.06	0.76 *	0.05	0.65 *	0.04	0.68 *	0.04

Note: λ = standardized factor loadings; SE = standardized error; composite reliability coefficient is in task and ego for each sample; * *p* < 0.01.

**Table 3 ijerph-19-03251-t003:** Model goodness-of-fit indices for the TEOSQ.

Model	Sample	χ^2^	df	χ^2^/df	B-S p	SRMR	CFI	TLI	RMSEA	90% CI
Previous studies											
Chi and Duda (1995) [37]	Spain 21.4 ± 1.3	108.21	62	1.74	<0.001	0.06	0.92	-	-	-
Franco et al. (2019) [14]	Spain 14.02 ± 1.19	503.05	103	4.88	<0.001	-	-	-	0.07	0.052–0.080
	Spain 20.3 ± 1.8	253.02	62	4.08	<0.001	0.08	0.83	-	-	-
López-Walle et al. (2011a) [15]	Mexico 13.8 ± 2.15	169.82	62	2.74	<0.001	-	0.91	-	0.09	-
Tomczak et al. (2020) [17]	Poland 19.2 ± 2.21	217.43	62	-	<0.001	0.05	0.95	0.93	0.06	0.053–0.071
Current study											
General		720.72	64	11.26	<0.001	0.06	0.94	0.93	0.07	0.062–0.071
Female		496.85	64	7.76	<0.001	0.06	0.92	0.93	0.07	0.065–0.077
Male		321.67	64	5.03	<0.001	0.06	0.94	0.95	0.07	0.058–0.072
Individual		525.26	64	8.21	<0.001	0.06	0.94	0.93	0.07	0.064–0.075
Team		306.01	64	4.78	<0.001	0.06	0.94	0.93	0.07	0.061–0.076

Note: - = data not provided by authors; χ^2^ = chi-square; df = degrees of freedom; χ^2^/df = normalized chi-square; B-S p = significance level Bollen—Stine Bootstrap (2000) samples; SRMR = Standardized Root Mean Square Residual; CFI = Comparative Fit Index; TLI = Tucker–Lewis Index; RMSEA = Root Mean Square Error of Approximation; CI = confidence interval.

**Table 4 ijerph-19-03251-t004:** Goodness-of-fit indices of invariance measures by gender and sport type for TEOSQ.

M	χ^2^	Df	Δχ^2^	Δdf	*p*	CFI	ΔCFI	SRMR	ΔSRMR	RMSEA	ΔRMSEA
Male—female											
CI	818.95	128	-	-	<0.001	0.94	-	0.06	-	0.05	-
MY	832.64	139	13.69	11	<0.001	0.94	0.00	0.06	0.00	0.05	0.00
SI	855.70	142	36.75	14	<0.001	0.94	0.00	0.07	0.01	0.05	0.00
RI	931.86	155	112.91	27	<0.001	0.93	0.01	0.06	0.00	0.05	0.00
Individual—team											
CI	831.27	128	-	-	<0.001	0.94	-	0.06	-	0.05	-
MY	860.82	139	29.55	11	<0.001	0.94	0.00	0.06	0.00	0.05	0.00
SI	900.04	142	68.77	14	<0.001	0.94	0.00	0.08	0.02	0.05	0.00
RI	966.02	155	134.75	27	<0.001	0.93	0.01	0.08	0.02	0.05	0.00

Note: M = Model; χ^2^ = chi—square; df = degrees of freedom; ∆χ^2^ = differences in chi-square values; ∆df = differences in degrees of freedom; CFI = Comparative Fit Index; ∆CFI = differences in Comparative Fit Index values; SRMR = Standardized Root Mean Square Residual; ∆SRMR = differences in Standardized Root Mean Square Residual values; RMSEA = Root Mean Square Error of Approximation; ∆RMSEA = differences in Root Mean Square Error of Approximation values; CI = configural invariance; MI = metric invariance; SI = scalar invariance; RI = residual invariance.

## Data Availability

Data are available upon request from the authors.

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
