# Peer review of "Task and Ego Orientation in Sport Questionnaire (TEOSQ): Psychometric Properties in Its Digital Version"

_ijerph, 2022, doi:10.3390/ijerph19063251_

Round 1

Reviewer 1 Report

The manuscript is well presented and the procedures are clear. The major thing missing is the rationale as to why this information is needed. The TEOSQ is an established data collection tool with good psychometric properties. It is a simple, 13-item questionnaire with common wording and minimal completion instructions. Therefore, could the authors explain why they believe the psychometric properties of an online version would be dramatically different to the paper version? There is no evidence to suggest there would be or that these would be significant enough to warrant further analysis. In fact, this manuscript's results indicate that the psychometric properties are good.

A full rationale for the need of this study is required.

Author Response

Dear Reviewer:

We are grateful for your comments, and we tried to improve the manuscript with your suggestions. Thank you.

Your comments encouraged us to include a deeper explanation in our text (line 65-88). As we said, online versions differ of paper version and it could affect in reliability because of the lower control of variables in the evaluation process. In that way, online version reduces the social desirability sometimes. In that way, Psychometric Properties could change in one or another type of evaluation. Please see the attachment in lines 65 to 88 to see the text changed.

Reviewer 2 Report

Dear Authors,

Please, find attached.

Thanks.

Kind Regards

Author Response

Thank you very much for your comments. We hope we have adequately responded to your suggestions.

First, in relation to multiple comments on expressions and syntax, the text has been corrected and modified following your suggestions.

Abstract

Line 13: “The sample was 2320 users” please, make sure this is "true"! https://www.mdpi.com/1660-4601/17/10/3593 https://www.ncbi.nlm.nih.gov/pmc/articles/PMC7277153/

Thank you for your comment. Our study sample is real, it has been done through a platform that has a high user traffic.

Line 20-22: please, insert: To conclude, considering the results obtained, the model shows a good psychometric fit for the sample in its online format. Please insert: practical implications and limitations.

The text has been included and modified.

Introduction

Line 32: Please insert reference.

We revised the reference, and it is included in the text, cited and referenced as [6].

Line 38: please insert reference.

Reference [1] has been included.

Line 59: and the disadvantages/limitations of this strategy?

Some arguments have been included in the text, like “the lower control of variables in the evaluation” and “the way it affect reliability”.

Line 65: please, insert study of 2021.

Two studies have been included (Beierle at al., 2021; Lakshminarasimhappa, 2021).

Materials and methods

Line 95: N= population, n= sample, please, insert n.

We have included n in the text.

Line 121: and the Oviedo Convention.

The Oviedo Convention has been included.

Results

Line 164: Please, insert the CV and SD in all tables

The table has been modified and CV and SD have been included in it.

Line 186: figure 1 Please, see the numbers 10, 11, 12 and 13.

The figure has been changed.

Discussion

Line 208: Please, see the abstract "The objective of this study was to analyse the psychometric properties of the online TEOSQ by MenPas 1.0 (www.menpas.com). "

The discussion has been modified.

Institutional Review Board Statement

Line 126 and 285: Oviedo Convention.

The Oviedo Convention has been included.

References

Line 288: please, insert more two studies of 2021.

Three studies of 2021 and 2022 have been included.

Round 2

Reviewer 1 Report

Thank you for addressing the previous concerns. This revision has greatly enhanced the manuscript.

Author Response

Thanks for your assessment. Your work has allowed to improve the manuscript.

Reviewer 2 Report

Dear Authors,

Please:

  1. Materials and Methods

2.1. Participants

(…) The 82.2% of the total (N = 2320) had higher education (n = 1908)…

Please, insert (n=2320).

N=population / n= sample

Figure 1. Structure of the TEOSQ.

Poor-quality

Table 3

Please, insert : Chi and Duda (1995) 

Thanks.

Kind Regards

Author Response

Thank you for your comments, it allows us to improve our manuscript.

Materials and Methods: 2.1. Participants: (…) The 82.2% of the total (N = 2320) had higher education (n = 1908)… Please, insert (n=2320). N=population / n= sample

The text has been changed.

Figure 1. Structure of the TEOSQ. Poor-quality.

The picture has been changed. Hope we have improve the quality.

Table 3. Please, insert: Chi and Duda (1995).

The text has been changed.

Thank you.